# Silicon Application Promotes Productivity, Silicon Accumulation and Upregulates Silicon Transporter Gene Expression in Rice

**DOI:** 10.3390/plants11070989

**Published:** 2022-04-05

**Authors:** Nanthana Chaiwong, Tonapha Pusadee, Sansanee Jamjod, Chanakan Prom-u-thai

**Affiliations:** 1Division of Agronomy, Department of Plant and Soil Sciences, Faculty of Agriculture, Chiang Mai University, Chiang Mai 50200, Thailand; nantana.c189@gmail.com (N.C.); tonapha.p@cmu.ac.th (T.P.); sansanee.j@cmu.ac.th (S.J.); 2Lanna Rice Research Center, Chiang Mai University, Chiang Mai 50200, Thailand

**Keywords:** rice, Si concentration, Si fertilizer, grain yield, *OsLsi6*

## Abstract

Rice has been shown to respond positively to Si fertilizer in terms of growth and productivity. The objective of this study was to evaluate the effect of a series of Si application rates on grain yield, Si concentration, and the expression of the *OsLsi6* gene among three Thai rice varieties. The varieties CNT1, PTT1, and KDML105 were grown in a pot experiment under six levels of Si (0, 100, 150, 200, 250, and 300 kg Si/ha). Grain yield was the highest at 300 kg Si/ha, being increased by 35%, 53%, and 69% in CNT1, PTT1, and KDML105, respectively, compared with the plants grown without added Si. For Si concentrations in rice plants, rising Si fertilizer application up to 150 kg/ha significantly increased the Si concentration in straw, flag leaf, and husk in all varieties. The Si concentration in all tissues was higher under high Si (300 kg Si/ha). Applying Si fertilizer also increased the expression level of *OsLsi6* in both CNT1 and PTT1 varieties. The highest expression level of *OsLsi6* was associated with 300 kg Si/ha, being increased by 548% in CNT1 and 326% in PTT1 compared with untreated plants. These results indicate that Si application is an effective way to improve rice yield as well as Si concentration, and that the effect is related to the higher expression of the *OsLsi6* gene.

## 1. Introduction

Rice is the world’s dominant cereal crop and staple food, and the demand is steadily increasing due to the rapidly growing population [1]. Therefore, rice production needs to be increased to satisfy consumer demand, but farmers are facing challenges such as climate change, the spread of plant diseases and pests, and decreasing rice-growing areas. Nutrient management has been suggested as an effective solution to increase rice productivity. However, applying only macronutrients (N, P, and K) fertilizer to the soil for improving plant growth and productivity may not be adequate for intensive rice cultivation. In addition, long periods of intensive rice cultivation and regular removal of rice straw every cropping season reduce the plant-available nutrient pools such as N, P, K, and especially Si [2,3]. Silicon (Si) is not identified as an essential element for the growth and production of plants, but it is well-known as a beneficial element, especially in rice, that can aid in alleviating both biotic and abiotic stresses [4,5]. Rice is a highly Si accumulating plant that shows a positive response to Si fertilizer regarding productivity [6]. Several studies have indicated that a sufficient Si supply increases rice grain yield by enhancing the number of panicles per plant as well as the number of spikelets and the percentage of filled grains [7,8]. Additionally, grain yield increases by Si application have been found to be associated with Si concentration in the shoot and grain [9]. A high concentration of Si in plant tissues has been demonstrated to play a major role in the growth and productivity of rice through increases in the strength of stems, the erectness of leaves, and the photosynthetic rate [10]. The wide variation of Si concentration in plants depends on the potential of uptake and transport mechanisms [11]. Generally, Si is taken up by rice roots in the form of silicic acid (SiOH_4_) via Si influx (Lsi1) and efflux transporters (Lsi2) [12,13,14]. Then, Si in solution in the soil is taken up by plant roots and translocated into the shoots through the transpiration stream in the xylem and is unloaded by Lsi6 [13,15,16]. The distribution of Si into the rice panicles and husk requires the expression of Lsi6; it has been reported that the knockout of the transporter gene resulted in decreasing Si accumulation in the panicles [16]. Our previous study [17] reported that Lsi6 is one of the key Si transporters that can be employed to improve rice production by increasing caryopsis development. However, limited information concerning Si treatment is available, especially regarding the application of Si fertilizer and the effects on grain yield, Si accumulation in different plant parts, and Si transporter gene expression among rice varieties. We hypothesized that grain yield and Si accumulation as well as Si transporter gene expression would vary under a series of Si application rates and rice varieties. Therefore, this study evaluated the effect of Si application rates on grain yield, Si concentration, and the expression of the *OsLsi6* gene among three Thai rice varieties selected based on their husk Si concentration. Thus, understanding the responses of rice varieties to Si fertilizer application on grain yield and Si concentration in rice plants from both physiological and molecular aspects will be useful for improving rice productivity through the management of Si fertilizer and breeding programs for high-Si rice.

## 2. Results

### 2.1. Grain Yield, Straw Dry Weight, and Yield Components

Si fertilizer affected grain yield and straw dry weight differently depending on the variety (*p* < 0.001) (Figure 1A,B, Table 1). Increasing Si fertilizer rates significantly increased grain yield and straw dry weight in all varieties compared with the plants grown without Si application (Figure 1A). Without the application of Si fertilizer, all varieties had the lowest grain yield, producing 33.3 g/plant in CNT1, 35.8 g/plant in PTT1, and 35.0 g/plant in KDML105. Applying Si fertilizer at 200 and 300 kg/ha increased grain yield in CNT 1 variety by 34% and 35%, respectively, in comparison with the control plants. In PTT1, increasing Si fertilizer at 100, 150, 200, and 300 kg/ha increased grain yield by 20%, 22%, 28%, and 53%, respectively, compared to the control plants, values that were not significantly different from plants treated with 100, 150, and 200 kg/ha. In KDML105, increasing Si fertilizer at 100, 150, 200, and 300 kg/ha increased grain yield by 39%, 43%, 64%, and 69%, respectively; however, there was no significant difference between 200 and 300 kg/ha doses. The highest grain yield was found at 300 kg Si/ha in all varieties, and the effect was stronger in KDML105 than in the other varieties. Similar to grain yield, the straw dry weight in all varieties increased with increasing Si fertilizer rates (Figure 1B). The greatest increases in straw dry weight (64.3% in CNT1, 52.5% in PTT1, and 33.4% in KDML105) occurred with the application of Si at 300 kg/ha compared to the control and the other treatments, and KDML105 had the highest straw dry weight compared with the other varieties.

#### Yield

Yield components were significantly influenced by Si application (*p* < 0.05) and varieties in all characters (*p* < 0.05), but without interaction between Si application and varieties (Table 2). Compared to the control treatment, the numbers of tillers per plant and panicles per plant were the highest at 300 kg/ha, being increased by 44% and 45%, respectively. However, these traits were similar among 100, 150, 200, and 300 kg/ha rates, and CNT1 had the highest numbers of tillers per plant and panicles per plant compared with the other varieties. Increasing Si rates increased the percent filled grain and 1000-grain weight in all varieties. The percent filled grain and 1000-grain weight were the highest at 300 kg/ha, being increased by 60% and 30%, respectively, compared with the plants grown without Si application, and there was no significant difference between 200 and 300 kg/ha doses in these components. Among the varieties, the highest percent filled grain and thousand-grain weight was found in KDML105 followed by PTT1, whereas CNT1 had the lowest percentage of filled grains and thousand-grain weight.

### 2.2. Silicon Concentration in Different Plant Parts

The Si concentrations in straw, flag leaf, and husk were affected by the Si fertilizer rate differently among the rice varieties (*p* < 0.05) (Figure 2A–C, Table 3). The lowest Si concentration in different plant tissues was found when rice plants were grown without Si application. Flag leaf and straw Si concentration increased with increasing Si up to 150 kg/ha in all varieties, whereas Si application at 50 and 100 kg/ha was not significantly different from the control plants. The highest Si application rate (300 kg/ha) increased flag leaf Si concentration by 50% in CNT1, 100% in PTT1, and 40% in KDML105; however, flag leaf Si concentration was not statistically different among 150, 200, and 300 kg/ha doses in CNT1 or KDML105 (Figure 2A). The highest husk Si concentration was increased by 132% in CNT1, 39% in PTT1, and 31% in KDML105 at 300 kg/ha compared with control plants. However, husk Si concentration was similar among 150, 200, and 300 kg/ha rates in PTT1 and KDML105, and between 200 and 300 kg/ha rates in CNT1 (Figure 2B). In addition, the maximum straw Si concentrations were 82% in CNT1, 32% in PTT1, and 116% in KDML105 at the highest Si fertilizer rate (300 kg/ha) compared with control plants (Figure 2C).

Significant correlations were found between grain yield and flag leaf Si concentration (r = 0.66, *p* < 0.01) and between grain yield and husk Si concentration (r = 0.48, *p* < 0.05). There was a positive correlation between the straw dry weight and flag leaf Si concentration (r = 0.91, *p* < 0.01). In addition, the percentage of filled grains was closely correlated with those of the flag leaf Si (r = 0.65, *p* < 0.01) and husk Si (r = 0.60, *p* < 0.01) (Table 4).

### 2.3. Total Silicon Uptake

The total Si uptake was significantly affected by Si fertilizer among varieties (*p* < 0.01) (Figure 3). Compared to the control treatment, applying Si fertilizer at 100, 150, 200, and 300 kg/ha increased the total Si uptake in CNT1 by 25%, 63%, 107%, and 137%, respectively, but 50 kg Si/ha had no effect, whereas 150, 200, and 300 kg/ha rates increased the total Si uptake in PTT1 by 33%, 68%, and 99%, respectively, and in KDML105, by 79, 89%, and 178%, respectively. The highest total uptake was obtained at 300 kg/ha in all varieties, stronger in KDML105 than in the other varieties.

### 2.4. Expression of OsLsi6 Gene

At flowering stages, the expression level of *OsLsi6* in the nodes of rice plants was significantly affected by Si application between rice varieties (Figure 4). Increasing the Si fertilizer rate significantly increased the expression level of *OsLsi6.* The highest expression level of *OsLsi6* was obtained at the 300 kg Si/ha rate, being increased by 548% in CNT1 and 326% in PTT1 compared with the expression of plants without the application of Si. Interestingly, CNT1 had a higher expression level of *OsLsi6* than PTT1 when Si was applied up to 100 kg/ha, whereas there was no significant difference between the varieties when gown under no Si. In addition, there were positive correlations between husk Si concentration and relative gene expression of *OsLsi6* in CNT1(r = 0.84, *p* < 0.01) and PTT1 (r = 0.60, *p* < 0.05) (Figure 5B). The straw Si concentration was significantly and positively correlated with the relative gene expression of *OsLsi6* in the PTT1 variety (r = 0.84, *p* < 0.01), but not in the CNT1 variety (Figure 5C). However, there were no significant correlations between flag leaf Si concentration and expression level of *OsLsi6* in any of the varieties (Figure 5A). Additionally, the total Si uptake was positively correlated with relative gene expression of *OsLsi6* in CNT (r = 0.68, *p* < 0.05) and PTT1 (r = 0.76, *p* < 0.01) (Figure 5D).

This study has shown that grain yield, Si concentration, total Si uptake, and the expression of a Si transporter gene responded differently to Si application rates among the rice varieties. Grain yield of all rice varieties was increased with the Si fertilizer rate, as reported in a previous study [18]. The improvement of rice crops by application of Si fertilizer was due to the increase in the yield characteristics of rice, as found in the present study where the numbers of tillers and panicles per plant and percentage of filled grains were increased under higher Si application [19]. Several studies have demonstrated that a high rate of Si fertilizer could promote the strength of stems, increase chlorophyll content and photosynthetic activity, and consequently improve productivity in rice crops [7,8,10]. The increase in rice grain yield by Si is also associated with an improvement in some morphological and physiological characteristics of the reproductive system, including pollen vigor, strength and thickness of the anther wall, and pollen germination rate [20]. Recent studies have reported that applying Si significantly enhances N and P content in the grain and straw of rice, resulting in higher dry weight and yield compared with no Si fertilizer [21,22]. Applying Si can increase the availability of P in the soil under P deficiency conditions by increasing soil pH and decreasing the availability of Fe and Mn, thereby indirectly improving P utilization by plants [23]. Under P sufficient soil, the straw dry weight and grain yield of rice did not respond to Si fertilizer application [24]. In this study, an increase in rice grain yield and straw dry weight may not be related to the concentration of P in the soil as the high extractable P (Bray II) in the soil at 35.04 mg/kg was found, which was above the critical level of soil P deficiency at 13 mg/kg [25]. Additionally, previous studies reported that Si fertilizer in the form of calcium silicate or sodium silicate increased soil pH which was consequently improved the availability of several plant-essential nutrients in the soil [26]. Grain yield improvement in this study may probably influence by calcium, which was not evaluated in comparison with the control treatment (no calcium silicate). Therefore, the results should be further confirmed by investigating the concentration of macro and micro-nutrients in both soil and plant tissues to distinguish the direct and indirect effect of Si fertilizer application in rice. This study has further demonstrated that all rice verities tested increased their grain yield under increasing Si application rates. Applying Si at 100 kg Si/ha potentially increased the grain yield of PTT1 and KDML105, whereas the application of 200 kg Si/ha was sufficient to enhance grain yield of the CNT1 variety. The highest grain yield was obtained with the highest level of Si fertilizer (300 kg Si/ha) in all varieties, stronger in KDML105 than in CNT and PTT1 varieties. This variation in grain yield response among varieties may be explained by genetic differences in the potential for Si uptake and the ability of rice plants to extract Si from the soil at low and high rates of Si application [27,28]. Although a higher application rate (300 kg Si/ha) strongly improved crop yield, the grain yield was similar among 200 and 300 kg Si/ha treatments. It can be concluded that the application of Si up to 200 kg Si/ha would be sufficient to enhance the sustainable production of Thai rice varieties, though this should be confirmed by increasing the number of rice varieties. Previous research has recommended that the application of Si at 225 kg Si/ha is the most economical measure for intensively irrigated paddy fields in tropical countries [29,30]. Additionally, grain yield was increased when Si fertilizer was applied at 450 kg/ha in rice crops, while it was slightly decreased when increasing *Si* to 600 kg/ha [31]. In China, applying Si at 600 kg/ha was found to decrease straw dry weight and grain yield in Si-enriched paddy soil [32]. Thus, the appropriate rate of Si fertilizer for rice crops can be varied according to rice variety and soil fertility conditions. The specific response at each cultivation area should be confirmed by varying Si rates and rice varieties to obtain accurate information.

In addition, applying Si fertilizer not only influences plant growth and productivity but also affects Si concentration in different parts of rice plants that influence productivity. This study found that the Si concentrations in flag leaf, straw, and husk in the three varieties varied depending on the Si rate, and the magnitude of the effect increased with the higher application of Si (300 kg/ha). This conclusion is supported by the strong positive correlation between the Si application rates and Si concentration in the rice plants. Previous studies have clearly demonstrated that the increased Si concentration resulting from increasing the Si application rate is presumably related to the increased Si availability in the soil that can enhance the function of the root system and help to stimulate the plant to absorb more Si from the soil [33,34]. Jinger et al. [35] reported that application of Si from 300 to 600 kg/ha was sufficient to increase Si in rice straw under Si-deficient soil, whereas Si concentration in the straw of rice increased by applying Si fertilizer up to 150 kg Si/ha in Si-sufficient soil. However, the need for Si fertilization depends on the rice variety, management practices, and the level of plant-available Si in the soil.

Genotypic variation in the effect of Si application rates on the concentration of Si in different parts of rice plants was observed in this study. In flag leaf, the most responsive variety to Si was PTT1, and CNT1 and KDML105 were moderately responsive. The response of husk Si to Si fertilizer was highly responsive in CNT1. In straw, KDML105 was much more responsive to Si fertilizer than the other varieties. This result suggests that some rice varieties accumulate high Si concentrations in plant tissues when grown under limited Si conditions. The responsive varieties are more desirable due to these varieties responding well to low as well as high Si rates. In addition, the moderately responsive varieties can be recommended for use under high Si application rates. The variation in response could be explained by the efficiency of Si uptake, transport, and distribution among rice varieties [27]. Previous studies reported that the architecture of the root system affected the Si uptake mechanism in rice plants [12,36,37]. Plants with more highly developed root systems would be able to extract more Si from the soil [38]. However, the parameter of the root was not examined in this study due to the growing conditions in the soil.

In addition, the present study has confirmed that high Si concentration in different plant parts of rice resulted in higher grain yield, indicating the significant correlations between grain yield and Si concentration in the husk and flag leaf. This finding was also in agreement with those of several studies demonstrating that grain yield increase was related to Si concentration in the shoot and grain through promoting the strength of stems, enhancing resistance to pathogens and insect pests, and increasing the erectness of leaves, thereby increasing the photosynthetic rate and consequently improving productivity [39]. In addition, Tamai and Ma [9] reported that the higher Si in the husk could increase grain *filling* by decreasing excessive transpiration to maintain the high moisture condition within the husk for normal caryopsis development. Thus, this study has established that increasing Si concentration in rice plants by Si application may be a key factor in improving rice productivity.

Additionally, the present study highlighted the finding that the total Si uptake in the whole rice plants responded differently to increasing Si rates among the varieties. The variety KDML105 was more responsive to Si application than CNT1 and PTT1, even though the concentration of Si in plant tissues was lower than in the other varieties. This variation in Si applied and Si total uptake responses among varieties may be attributed to the varietal characteristics; KDML105 is a traditional rice variety with a tall plant type, whereas CNT1 and PTT1 are modern rice varieties with a semi-dwarf plant type. The rice varieties may specifically respond to Si application in their Si uptake and accumulation, as has been reported in a previous study [40]. However, further work is necessary to understand how different rice varieties respond to Si fertilizer rates, especially in comparisons between the modern rice variety and traditional rice variety groups.

Another possible reason for the differences in Si uptake, transportation, and distribution in rice shoots among varieties may be related to the expression levels of Si transporter genes. In rice, it is well-established that several genes, including *OsLsi1, OsLsi2,* and *OsLsi6*, are major genes controlling Si uptake and transportation [11,13,15]. The results from gene expression analysis revealed that the expression of *OsLsi6* in the node of CNT1 and PTT1 at the flowering stage responded to the Si application rate. The expression level of *OsLsi6* was increased with rising Si fertilizer rates in both varieties, more strongly in CNT1 than in PTT1, in agreement with higher Si concentration in CNT1 than PTT1 in all plant parts except for the flag leaf. This result is in accordance with a report in maize that showed that further increasing soil Si fertilizer strongly increased the expression level of the *ZmLsi6* gene in leaves [41]. In rice, applying Si fertilizer at the reproductive stage was found to increase the expression leave of *OsLsi6* [42], whereas the opposite expression of *OsLsi6* was found in the root and leaf blades during the vegetative growth stage [13]. Thus, the expression level response of Lsi6 to Si supply can be varied by many factors, e.g., rice variety, plant part, and growth stage. This study has demonstrated that Si concentration in rice plants and Si total uptake were related to gene expression of *OsLsi6*, as indicated by the positive correlation between the expression of the *OsLsi6* gene and the Si concentration in straw and husk and between gene expression of *OsLsi6* and Si total uptake. This was confirmed by the recent study of Yamaji and Ma [15] who reported that the knockout of Lsi6 resulted in decreasing Si deposition in the leaf blades, leaf sheaths, and panicles but increased Si in the flag leaf. In addition, Lavinsky et al. [42] found that a high level of Si in the panicle is required for up-regulating the *OsLsi6* expression. Thus, the association between Si concentration in rice plants and *OsLsi6* gene expression may be the key factor for increasing Si concentration as well as Si total uptake in rice plants, resulting in improving grain yield. Therefore, insufficient Si fertilizer can not only decrease Si accumulation and the expression of the *OsLsi6* gene in rice plants but could also reduce rice productivity.

Authors should discuss the results and how they can be interpreted from the perspective of previous studies and the working hypotheses. The findings and their implications should be discussed in the broadest context possible. Future research directions may also be highlighted.

## 3. Materials and Methods

### 3.1. Plant Culture

The experiment was conducted under greenhouse conditions during the rainy season (June to September) in 2018 at the Faculty of Agriculture, Chiang Mai University, Thailand. The experiment was arranged in a factorial completely randomized design with three independent replications. The three modern rice varieties used in this experiment were Chainat 1 (CNT1), Pathumthani 1 (PTT1), and Khao Dawk Mali 105 (KDML105), selected with respect to their husk Si concentration, with 8.7%, 7.1%, and 6.7% dry weight of husk Si concentration, respectively [43]. Plants were grown in 15 cm-diameter pots containing 4 kg soil. The chemical profile of the soil used in the experiments was the Sansai series, a soil with a sandy loam texture. Soil pH 6.4 (1:1, soil:water), organic matter 1.38% (Walkley–Black method); total N 0.07% (Kjeldahl method); available phosphorus 35.06 mg/kg (Bray II), exchangeable potassium 39.87 mg/kg (NH_4_OAc, pH 7), and extractable Zn 0.73 mg/g (DTPA). Si was applied in the form of calcium silicate (Ca_2_SiO_4_) at six levels (0, 50, 100, 150, 200, and 300 kg/ha) seven days after transplanting. The seeds were pre-germinated by soaking in distilled water for 24 h and incubated in moistened plastic cups for two weeks. The seedlings were transplanted into the prepared pots, one plant per pot. The fertilizer was applied using 15N-15P-15K at 0.25 g per pot, splitting into four times at seven days after transplanting, tillering, booting, and flowering stages.

### 3.2. Data Collection and Sample Preparation

At maturity, plant samples were harvested and evaluated for grain yield, straw dry weight, and yield components (number of tillers and panicles per plant, percent filled grains, and 1000 grain weight). The samples were separated into the straw, flag leaf, and husk and analyzed for Si concentration. The collected samples were washed twice with distilled water before being oven-dried at 75 °C for 48 h. The dried samples were mechanically ground in a hammer mill for Si concentration analysis.

### 3.3. Silicon Concentration Analysis

The Si concentration was determined by the method of Dai et al. [44] with modification. Approximately 10 g of dried samples were ground, and 0.1 g of subsamples were extracted with 3 mL of 50% NaOH, then autoclaved at 121 °C for 20 min. The extract was transferred into a volumetric flask and adjusted to 50 mL with ddH_2_O, and then the liquid was filtered through Whatman No. 1 filter paper. The 1 mL sample solution was transferred to a 50 mL volumetric flask, and 30 mL 20% acetic acid and 10 mL ammonium molybdate solution (pH 7.0) were added. The mixtures were shaken and kept at room temperature for 5 min before adding 5 mL of 20% tartaric acid and 1 mL of reducing solution. After 30 min, the absorbance was measured with a spectrophotometer at 650 nm.

### 3.4. Gene Expression by Semi-Quantitative RT-PCR Analysis

The two modern rice varieties (Chainat 1 (CNT1) and Pathumthani 1 (PTT1)) were grown in soil culture with four rates of Si application (0, 100, 200, and 300 kg/ha). At flowering stages, the first nodes of plants were collected immediately frozen in liquid nitrogen. The total RNA was extracted from frozen node tissues using PureLinkTM RNA Mini Kit (Invitrogen, Thermo Fisher Scientific). RNA quality was checked according to Wangkaew et al. [45]. Total RNA was treated with DNaseI and 1 μg of total RNA was then pipetted for cDNA synthesis using a RevertAid first-strand cDNA synthesis Kit (Thermo scientific). Gene expression levels of *OsLsi6* were analyzed by semi-quantitative RT-PCR using gene-specific primers of *OsLsi6* and *OsActin* [46]. The primer sequences used were: *OsLsi6*, forward 5′-GAGTTCGACAACGTCTAATCGC-3′ and reverse 5′-AGTACACGGTACATGTATACACG-3′ [45]; *OsActin*, forward5′-GACTCTGGTGATGGTGTCAGC-3′ andreverse 5′-GGCTGGAAGAGGACCTCAGG-3′ [14]. PCR was conducted in triplicate for amplification of cDNA templates with *OsLsi6* and *OsActin1* with the following reaction: 2 µL of 1:20 diluted template cDNA in a total volume of 1 µg, 14 µL of water (ddH_2_O), 4 µL of 5× MyTaq Reaction Buffer, 0.2 µL of forward and reverse primer, 0.1 µL of 5 unit MyTaq TM HS DNA Polymerase (Bioline, UK), and 0.6 µL of DMSO (total volume 20)µL. Reaction conditions for thermal cycling were 95 °C for 2 min, followed by 40 cycles of 95 °C for 30 s, 55 °C for 30 s, then 72 °C for 30 s. The related transcript quantification was performed using relative intensity to the reference gene (*OsActin1*).

### 3.5. Statistical Analysis

Statistical analysis of all data was performed by using Statistic 9 (analytical software SX). Analysis of variance (ANOVA) was used to detect the difference among treatments, and the least significant difference (LSD) at *p* < 0.05 was used to compare means. The significance of correlations was analyzed by Pearson linear correlation. Gene expression levels were analyzed by relative intensity to the reference gene (Actin) using ImageJ software version 1.50i (Wayne Rasband, National Institutes of Health, Bethesda, MD, USA). The relative intensity of gene expression was subjected to statistical analysis using statistical software.

## 4. Conclusions

In conclusion, this study demonstrated that rice yield increased with increasing Si application rate and increased flag leaf, straw, and husk Si concentrations, and this was related to the higher expression level of the *OsLsi6* gene. The response of rice varieties in their Si accumulation in different plant parts and grain yield should be verified under diverse cultural management methods. Identifying the response of rice varieties in grain yield and Si concentration under various cultural management strategies would provide useful information for selection and management to achieve higher yields. Therefore, applying Si fertilizer is an effective practice for improving Si concentration in plant parts and consequently improving grain yield in rice crops. However, the effect of Si fertilizer application on Si accumulation in different plant parts and grain yield should be further examined for confirmation by increasing the number of rice varieties. In addition, the other Si transporter genes in rice plants such as Lsi1 and Lsi2 deserve further investigation.

## Figures and Tables

**Figure 1 plants-11-00989-f001:**
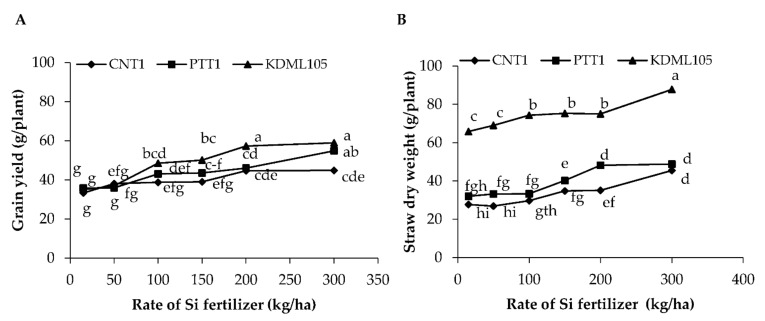
Grain yield (**A**) and straw dry weight (**B**) of three rice varieties (CNT1, PTT1 and KDML105) grown in soil culture with six rates of Si fertilizer (0, 50, 100, 150, 200 and 300 kg/ha) under waterlogged conditions. Different lowercase letters indicate significant differences at *p* < 0.05.

**Figure 2 plants-11-00989-f002:**
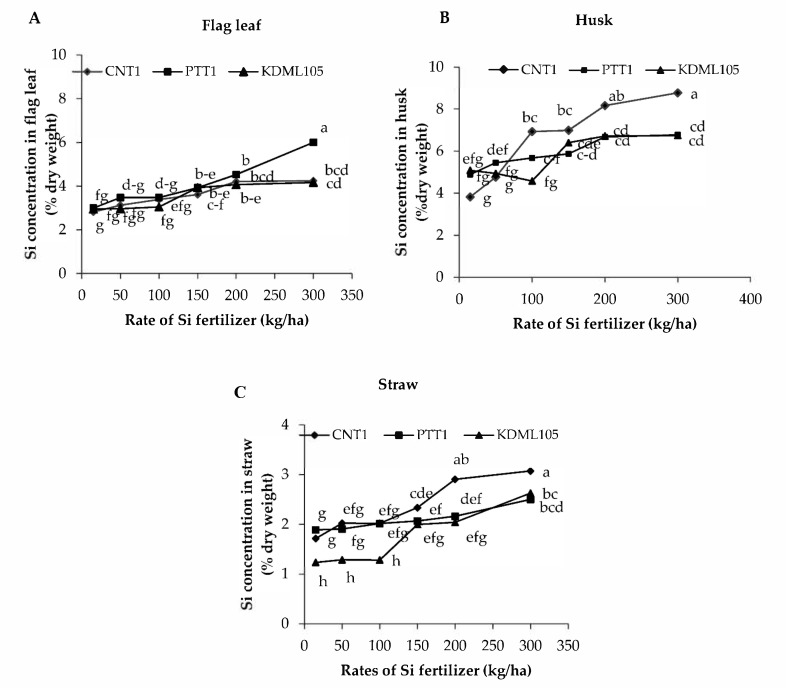
Silicon concentration in flag leaf (**A**), husk (**B**), and straw (**C**) of three rice varieties (CNT1, PTT1, and KDML105) grown in soil culture with six rates of Si fertilizer (0, 50, 100, 150, 200, and 300 kg/ha). The samples were harvested at maturity. Different lowercase letters indicate significant differences at *p* < 0.05.

**Figure 3 plants-11-00989-f003:**
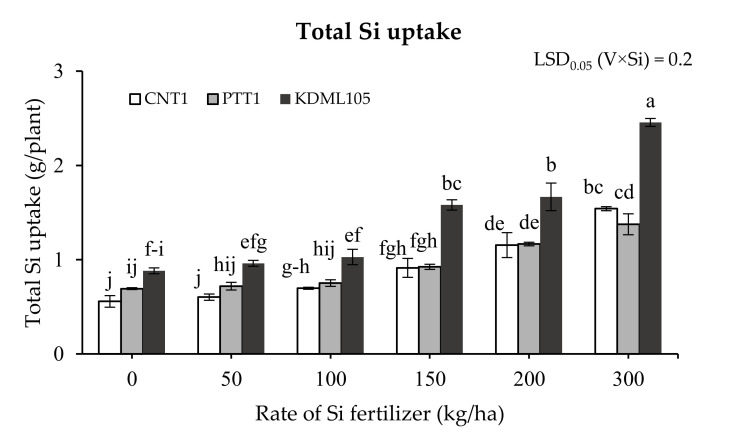
Total Si uptake of three rice varieties (CNT1, PTT1, and KDML105) grown in soil culture with six rates of Si fertilizer (0, 50, 100, 150, 200, and 300 kg Si/ha). The samples were harvested at maturity. Different lowercase letters indicate significant differences at *p* < 0.05. The error bars are the standard errors of each treatment mean from three replications.

**Figure 4 plants-11-00989-f004:**
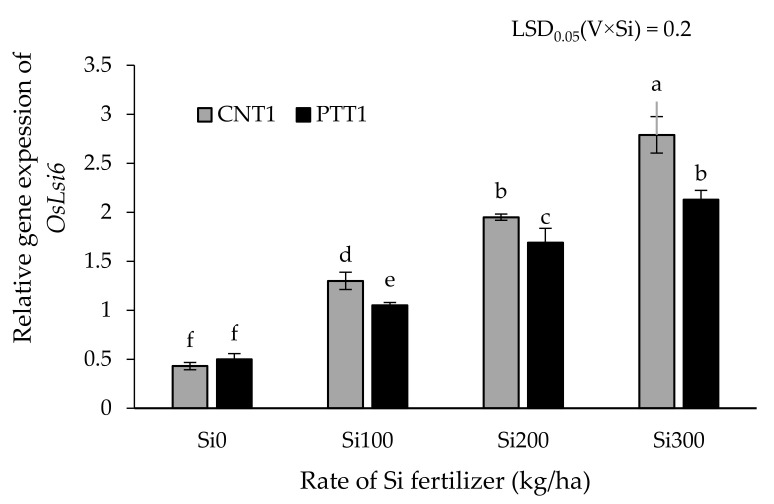
Relative gene expression of Si transport genes *OsLsi6* of two rice varieties (CNT1 and PTT1) at flowering stages grown in soil culture with four rates of Si application (0, 100, 200, and 300 kg/ha) by semi-quantitative RT-PCR analysis. Different lowercase letters indicate significant differences at *p* < 0.05. The error bars are the standard errors of each treatment mean from three replications.

**Figure 5 plants-11-00989-f005:**
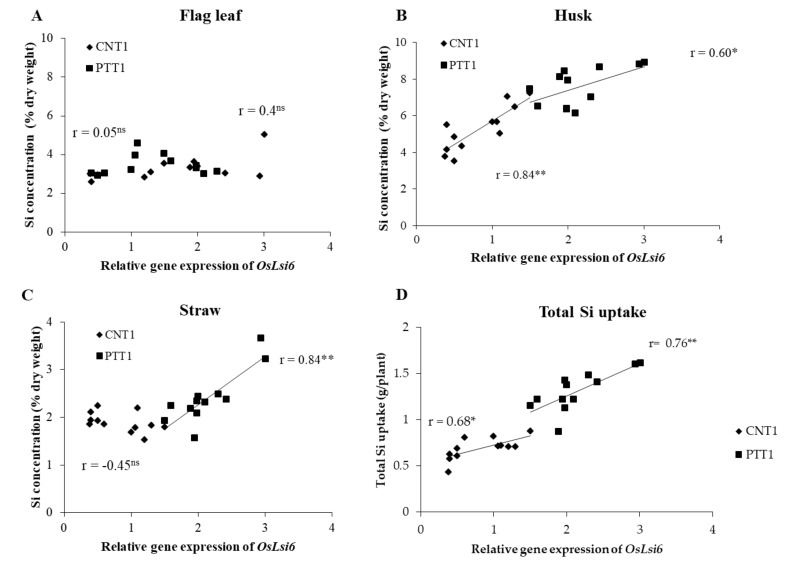
Correlation between relative gene expression of *OLsi6* and Si concentration in flag leaf (**A**), husk (**B**), and straw (**C**) and total Si uptake of two rice varieties (CNT1 and PTT1) (**D**) grown in soil culture with four rates of Si application (0, 100, 200, and 300 kg/ha). *, ** = significantly different at *p* < 0.05 and *p* < 0.01, respectively, and ns = non-significantly different at *p* < 0.05.

**Table 1 plants-11-00989-t001:** Analysis of variance (ANOVA) of grain yield and straw dry weight among three rice varieties grown in soil culture with six rates of Si fertilizer (0, 50, 100, 150, 200, and 300 kg/ha) under waterlogged conditions.

	Grain Yield	Straw Dry Weight
Variety (V)	*p* < 0.01	*p* < 0.01
Si application (Si)	*p* < 0.01	*p* < 0.01
V × Si	*p* < 0.05	*p* < 0.05
LSD_0.05_ (V × Si)	6.8	5.1

**Table 2 plants-11-00989-t002:** Yield components of three rice varieties (i.e., CNT1, PTT1, and KDML105) grown in soil with six rates of Si fertilizer application (0, 50, 100, 150, 200, and 300 kg/ha).

	Tiller/Plant	Panicle/Plant	Filled Grain (%)	1000 Grain Weight (g)
Si application rate (kg/ha)				
Si0	12.44 ^b^	12.44 ^c^	90.58 ^e^	28.78 ^d^
Si50	12.44 ^b^	12.59 ^c^	92.61 ^d^	30.59 ^cd^
Si100	15.89 ^a^	16.10 ^ab^	93.33 ^cd^	32.22 ^cd^
Si150	15.71 ^a^	15.56 ^ab^	94.31 ^bc^	32.90 ^bc^
Si200	16.89 ^a^	16.41 ^ab^	95.49 ^ab^	35.60 ^ab^
Si300	17.86 ^a^	18.00 ^a^	96.50 ^a^	36.81 ^a^
Rice variety				
CNT1	17.90 ^a^	17.49 ^a^	92.68 ^b^	30.00 ^b^
PTT1	15.21 ^b^	15.20 ^b^	93.11 ^b^	33.87 ^a^
KDML105	12.41 ^c^	12.90 ^c^	95.60 ^a^	30.00 ^b^
F-test				
Variety (V)	*p* < 0.01	*p* < 0.01	*p* < 0.01	*p* < 0.01
LSD_0.05_ (V)	1.5	1.5	11.4	2.4
Si application (Si)	*p* < 0.01	*p* < 0.01	*p* < 0.01	*p* < 0.01
LSD_0.05_ (Si)	2.1	2.1	16.2	3.3
V × Si	-	-	-	-

Different lowercase letters indicate significant differences at *p* < 0.05.

**Table 3 plants-11-00989-t003:** Analysis of variance (ANOVA) of Silicon concentration in flag leaf, husk, and straw among three rice varieties grown in soil culture with six rates of Si fertilizer (0, 50, 100, 150, 200, and 300 kg/ha) under waterlogged conditions.

	Si Concentration (% Dry Weight)			Total Si Uptake (g/Plant)
	Flag Leaf	Husk	Straw	
Variety (V)	*p* < 0.01	*p* < 0.01	*p* < 0.01	*p* < 0.01
Si application (Si)	*p* < 0.01	*p* < 0.01	*p* < 0.01	*p* < 0.01
V × Si	*p* < 0.05	*p* < 0.05	*p* < 0.05	*p* < 0.01
LSD_0.05_ (V × Si)	0.7	1.35	0.4	0.2

**Table 4 plants-11-00989-t004:** Correlations between silicon concentration in different agronomic traits, grain yield, and yield components of three rice varieties grown in soil culture with six rates of Si fertilizer (0, 50, 100, 150, 200, and 300 kg/ha) under waterlogged conditions. The concentration of Si in different agronomic traits was analyzed at the maturity stage.

	Si Concentration (% Dry Weight)		
	Flag Leaf	Husk	Straw
Grain yield (g/plant)	0.66 **	0.48 *	0.42 ^ns^
Straw dry weight (g/plant)	0.91 **	0.04 ^ns^	0.19 ^ns^
Filled grain (%)	0.65 **	0.60 ^**^	0.08 ^ns^
Number of tillers/plant	0.37 ^ns^	0.30 ^ns^	0.44 ^ns^
Number of panicles/plant	0.40 ^ns^	0.29 ^ns^	0.41 ^ns^

ns = not significant, * = significant at *p* < 0.05, ** = significant at *p* < 0.05.

## Data Availability

No new data were created or analyzed in this study. Data sharing is not applicable to this article.

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
