# Peer review of "Silicon Application Promotes Productivity, Silicon Accumulation and Upregulates Silicon Transporter Gene Expression in Rice"

_plants, 2022, doi:10.3390/plants11070989_

Round 1
Reviewer 1 Report
I really appreciate Editor to invite me to review the manuscript entitled “Silicon Application Promotes Productivity, Silicon Accumulation and Upregulate Silicon Transporter Gene Expression in Rice” written by Nanthana Chaiwong et al.
After a careful reading, this manuscript is quite interesting looking at the effects of silicon fertilization on the effect of rice growth and development with some solid data.
It is interesting to show the relationship between silicon fertilizer and gene expression in rice. It relatively well-written, and well-organized, and systematized.
It is worthy of publication in Plants after a moderate revision. yet I have some comments to be considered by authors.
At line 32-33, considering the topic to apply Si, the much prefer to refer as follows: the intensive rice cultivation, regular straw remove, and soil weathering substantially reduce the plant-available nutrient pools such as N, P, and K, especially Si. Thus, Si fertilizer supply may be required to sustain continuous rice cropping (ref. Li and Delvaux 2019 Gcb Bioenergy, 11(11), pp.1264-1282; Haynes RJ. Journal of Plant Nutrition and Soil Science. 2014 Dec;177(6):831-44; Puppe et al., Geoderma, 403, p.115187.).
Strange! Materials and Method should be at the front of Results? No?
Figure 2 should be re-worked to a better reading.
Author Response
Dear Editor,
Manuscript title "Silicon Application Promotes Productivity, Silicon Accumulation and Upregulate Silicon Transporter Gene Expression in Rice"
Manuscript ID: plants-1643685
The manuscript has been revised by the authors as in the following. Amendments to the manuscript are highlighted in yellow for reviewer 1, green for reviewer 2, blue for reviewer 3, and pink for reviewer 4
Reviewer 1
Comment
At line 32-33, considering the topic to apply Si, the much prefer to refer as follows: the intensive rice cultivation, regular straw remove, and soil weathering substantially reduce the plant-available nutrient pools such as N, P, and K, especially Si. Thus, Si fertilizer supply may be required to sustain continuous rice cropping (ref. Li and Delvaux 2019 Gcb Bioenergy, 11(11), pp.1264-1282; Haynes RJ. Journal of Plant Nutrition and Soil Science. 2014 Dec;177(6):831-44; Puppe et al., Geoderma, 403, p.115187.).
Response:
Thank you for your recommendation. We have added the suggested references in the introduction in lines 34-36.
Comment
Strange! Materials and Method should be at the front of Results? No?
Response:
This is the format of the journal.
Comment
Figure 2 should be re-worked to a better reading.
Response:
Figure 2 has been revised.
Reviewer 2 Report
In this manuscript, the authors evaluated the effect of a series of Si application rates on grain yield, Si concentration, and the expression of the OsLsi6 gene among three Thai rice varieties. Here are some questions about the manuscript:
- Three rice varieties were used in the manuscript, why do the authors select them for analysis? Are they representative materials for different rice groups?
- Why don’t the authors conduct semi-quantitative RT-PCR analysis for Khao Dawk Mali 105?
- The authors mentioned Si was applied seven days after transplanting, but the expression levels of OsLsi6 were analyzed by using the first nodes of plants at flowering stages. Is there any relationship between OsLsi6 expression level and Si supply?
- It is not clear which method was used for gene expression analysis.
- Four levels of Si were used for analysis and grain yield was the highest at 300 kg Si/ha, it is better to detect higher Si level than 300.
- It’s better to use line chart for Fig1 and Fig2, and the table in Figure should be separated.
- When the authors compare the difference of grain yield, straw dry weight etc., it is better to normalize the data with the control without Si first, so we can see the effect of Si supply.
- Many references used in the manuscript were not correct. For example, the authors mentioned the demand of rice in reference 1, which talked about uptake system of silicon; none of the references from 2 to 5 focused on biotic or abiotic stress; Reference 10 analyzed aquaporin gene family in Cannabis sativa, which was not rice the authors talked in the manuscript.
Author Response
Dear Editor,
Manuscript title "Silicon Application Promotes Productivity, Silicon Accumulation and Upregulate Silicon Transporter Gene Expression in Rice"
Manuscript ID: plants-1643685
The manuscript has been revised by the authors as in the following. Amendments to the manuscript are highlighted in yellow for reviewer 1, green for reviewer 2, blue for reviewer 3, and pink for reviewer 4
Reviewer 2
In this manuscript, the authors evaluated the effect of a series of Si application rates on grain yield, Si concentration, and the expression of the OsLsi6 gene among three Thai rice varieties. Here are some questions about the manuscript:
Comment
Three rice varieties were used in the manuscript, why do the authors select them for analysis? Are they representative materials for different rice groups?
Response: The three rice varieties were selected to use in this study with respect to their husk Si concentration; namely high husk Si (CNT1), and low husk Si (PTT1, KDML105). The sentences were added in lines 61-62 and 370-373.
Comment
Why don’t the authors conduct a semi-quantitative RT-PCR analysis for Khao Dawk Mali 105? Response:
We have a problem during carrying on the experiment. The lack of gene expression result was due to the primers of the OsLsi6 gene could not amplify the DNA fragment from the cDNA of the KDML105 variety. However, the variety PTT1 is representative of a low Si husk concentration for this study.
Comment
The authors mentioned Si was applied seven days after transplanting, but the expression levels of OsLsi6 were analyzed by using the first nodes of plants at flowering stages. Is there any relationship between OsLsi6 expression level and Si supply?
Response:
This research has reported that applying Si fertilizer at the early growth stage could effectively increase soil available Si and soil pH leading to increased Si accumulation in plants through maturity, which was related to the level of OsLsi6 gene expression at the flowering stage.
Comment
It is not clear which method was used for gene expression analysis.
Response:
The gene expression analysis has been clearly mentioned in materials and methods in lines 403-409.
Comment
Four levels of Si were used for analysis and grain yield was the highest at 300 kg Si/ha, it is better to detect higher Si level than 300.
Response:
This is a great comment. We are carrying on an experiment to evaluate the response of grain yield and Si accumulation in rice under the higher Si fertilizer rate of over 300 kg/ha. The result will be present in future research.
Comment
It’s better to use line chart for Fig1 and Fig2, and the table in Figure should be separated.
Response:
Fig1 and Fig2 have been added to the line chart and the table in Figures have been separated.
Comment
When the authors compare the difference of grain yield, straw dry weight etc., it is better to normalize the data with the control without Si first, so we can see the effect of Si supply.
Response:
The results have been improved according to the reviewer’s comment.
Comment
Many references used in the manuscript were not correct. For example, the authors mentioned the demand of rice in reference 1, which talked about uptake system of silicon; none of the references from 2 to 5 focused on biotic or abiotic stress; Reference 10 analyzed aquaporin gene family in Cannabis sativa, which was not rice the authors talked in the manuscript.
Response:
The references have been checked and corrected throughout the manuscript.
Reviewer 3 Report
No comments.
What is the main question addressed by the research?
How Si application is affecting Si accumulation and productivity of rice.
Is it relevant and interesting?
This is very relevant and interesting as rice the main crop in world feed half of human population. How original is the topic?
The topic is original as it combines traditional measurements with measurements on gene expression of Si transporters.
What does it add to the subject area compared with other published material?
The manuscript adds a more comprehensive understanding on how Si application, gene expression of Si transporters and productivity are related together.
Is the text clear and easy to read?
The manuscript is easy to read.
Are the conclusions consistent with the evidence and arguments presented?
All conclusion made are consistent with the data presented.
Do they address the main question posed?
The manuscript addressed all the main questions properly.
Are the references appropriate?
The references of this manuscript are well chosen.
Any additional comments on the tables and figures.
The presentation of the data is good.
Author Response
Dear Editor,
Manuscript title "Silicon Application Promotes Productivity, Silicon Accumulation and Upregulate Silicon Transporter Gene Expression in Rice"
Manuscript ID: plants-1643685
The manuscript has been revised by the authors as in the following. Amendments to the manuscript are highlighted in yellow for reviewer 1, green for reviewer 2, blue for reviewer 3, and pink for reviewer 4
Reviewer 3
Thank you for all your valuable comments and suggestions.
Reviewer 4 Report
In this study, the authors show an interesting information that Si application can enhance rice yield and correlate with OsLsi6 gene expression. The experiments are fairly designed and analyzed. However, the authors need to provide more detailed information.
-
- In the results, P3, L86, P5, L132, the two tables do not have descriptions. It is recommended authors provided the information.
- In Table 1, the value and the decimal point must be aligned. The lowercase letter needs to be superscripted.
- P5, L134, L136, the description of ‘in different plant parts’, it is recommended to revise as ‘in different agronomic traits’.
- In Figure 4, why not use the rice material ‘KDML105’ to analyze the gene expression of OsLsi6?
- In Figure 5d, the linear regression did not seem like have a significant correlation between total Si uptake and expression level of OsLsi6. It is recommended authors recheck the date.
Author Response
Dear Editor,
Manuscript title "Silicon Application Promotes Productivity, Silicon Accumulation and Upregulate Silicon Transporter Gene Expression in Rice"
Manuscript ID: plants-1643685
The manuscript has been revised by the authors as in the following. Amendments to the manuscript are highlighted in yellow for reviewer 1, green for reviewer 2, blue for reviewer 3, and pink for reviewer 4
Reviewer 4
In this study, the authors show an interesting information that Si application can enhance rice yield and correlate with OsLsi6 gene expression. The experiments are fairly designed and analyzed. However, the authors need to provide more detailed information.
Comment
In the results, P3, L86, P5, L132, the two tables do not have descriptions. It is recommended authors provided the information.
Response:
The information has been added in the caption in Tables 1 and 2.
Comment
In Table 1, the value and the decimal point must be aligned. The lowercase letter needs to be superscripted.
Response:
The details in table 1 have been revised according to the reviewer’s comment.
Comment
P5, L134, L136, the description of ‘in different plant parts’, it is recommended to revise as ‘in different agronomic traits’.
Response:
This part has been revised according to the reviewer’s comment.
Comment
In Figure 4, why not use the rice material ‘KDML105’ to analyze the gene expression of OsLsi6?
Response:
We have a problem during carrying on the experiment. The lack of gene expression result was due to the primers of the OsLsi6 gene could not amplify the DNA fragment from the cDNA of the KDML105 variety. However, the variety PTT1 is representative of a low Si husk concentration for this study.
Comment
In Figure 5d, the linear regression did not seem like have a significant correlation between total Si uptake and expression level of OsLsi6. It is recommended authors recheck the date.
Response:
Figure 5d has been carefully checked and revised.
Round 2
Reviewer 2 Report
The comments have been responsed.